# University–Industry Technology Transfer: Empirical Findings from Chinese Industrial Firms

**Jiaming Jiang** [1,*] **, Yu Zhao** [2] **and Junshi Feng** [1]

1   Graduate School of Humanities and Social Science, Okayama University, 3-1-1 Tsushimanaka, Kitaku, Okayama 700-8530, Japan
2   School of Management, Department of Management, Tokyo University of Science, 1-3 Kagurazaka, Shinjuku-ku, Tokyo 162-8601, Japan
*   Correspondence: jiaming@okayama-u.ac.jp

**Abstract:** The knowledge and innovation generated by researchers at universities is transferred to industries through patent licensing, leading to the commercialization of academic output. In order to investigate the development of Chinese university–industry technology transfer and whether this kind of collaboration may affect a firm's innovation output, we collected approximately 6400 license contracts made between more than 4000 Chinese firms and 300 Chinese universities for the period between 2009 and 2014. This is the first study on Chinese university–industry knowledge transfer using a bipartite social network analysis (SNA) method, which emphasizes centrality estimates. We are able to investigate empirically how patent license transfer behavior may affect each firm's innovative output by allocating a centrality score to each firm in the university–firm technology transfer network. We elucidate the academic–industry knowledge by visualizing flow patterns for different regions with the SNA tool, Gephi. We find that innovation capabilities, R&D resources, and technology transfer performance all vary across China, and that patent licensing networks present clear small-world phenomena. We also highlight the Bipartite Graph Reinforcement Model (BGRM) and BiRank centrality in the bipartite network. Our empirical results reveal that firms with high BGRM and BiRank centrality scores, long history, and fewer employees have greater innovative output.

**Keywords:** collaborative networks; technology transfer; China; university–firm collaboration; social network analysis; economic policy; economic statistics

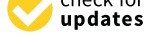



## 1. Introduction

The transfer of knowledge and technology from public universities to private sectors has attracted attention in academic research. Besides teaching and research, academic institutes engage in a 'third mission' through knowledge and technology transfer activities [1–3]. It is commonly accepted that universities are unique actors in the production and delivery of new knowledge that supports economic development [4–7]. However, to play an important role in the economy, it is necessary for new knowledge not only to be created at universities, but also to be transferred from universities to society, or more precisely, to industries [8]. There are various channels of knowledge transfer [9–11], such as publications, conferences, consulting, licensing, joint ventures, personnel exchange, and scientist migration to private sectors [12]. Through these channels, patenting and licensing has attracted the most attention in both legislative practices and academic research, and licensing is considered one of the crucial ways for universities to transfer scientific knowledge [13]. The knowledge and innovation generated by researchers at universities is transferred to industries through patent licensing, thus bringing academic output much closer to the stage of commercialization. Additionally, private firms may intend to pursue more profits through acquiring new technologies. Licensing contracts with universities can have firms pay royalties for technologies that can be applied in the production process

to make profits [12]. On the other hand, recent developments in the field of social network analysis (SNA) have resulted in software tools for visualization as well as improved analysis and interpretation of patent statistics [14]. With the acceleration of network processes, the mode of knowledge transfer is gradually transforming from linear to network mode [15]. Social networks are increasingly considered to be influential in explaining the knowledge transfer process. Using the network framework of understanding, knowledge transfer refers to the effort of a source to share knowledge with a receiver and the receiver's effort to acquire and to absorb this knowledge [16]. Each member of the network acts as a source of knowledge itself by providing knowledge to another member, and/or acts as a conduit through which other members can access knowledge [17]. From this perspective, licenses serve as conduits for the transfer of information and knowledge. Furthermore, the network structure of a hyperlinked environment can be a rich source of information about the content of the environment, provided we have effective means for understanding such a structure [18]. Among the ways of surveying the structure of a network, centrality is a key structural feature of social networks and communicates important information about an individual's prominence or role within a given network [19,20]. In this paper, we apply the output from two bipartite centrality algorithms, BGRM and BiRank, to a real-world bipartite social network of university–firm knowledge transfer processes and contribute to elucidating the following question: how might the centrality estimates in a bipartite knowledge transfer network affect a firm's innovation output? Further, what tangible benefits have been brought to the development of the innovation of industrial firms through such research?

So far, limited research has been conducted on knowledge flow networks or technology transfer networks within a single country [21]. While the literature has mainly focused on developed countries as an empirical setting, we lack understanding on the evolution of knowledge transfer networks in developing countries [16]. In this paper, we focus on China—the world's second largest economy.

We employed a bipartite SNA technique to empirically analyze the characteristics of firms in a university–industry knowledge transfer network in China. The main contributions of the present study are as follows: We employed important methods for exploring bipartite networks, emphasizing the visual exploration of the knowledge and technology transfer from public universities to firms. Second, we utilized patent applications as output indicators for R&D investment and combined the characteristics of firms with their centrality score to investigate the effects of these characteristics on the firms' innovative output. This is the first study on Chinese university–firm knowledge transfer using a bipartite network analysis method that emphasizes the centrality index.

## 2. Literature Review and Theoretical Background

### 2.1. Background of Chinese University–Industry Technology Transfer

China has made great progress along the road of independent innovation, research, and development investment, and the number of academic achievements and patents has been ranked as the top in the world [22]. By improving the performance of Chinese university knowledge and technology transfer through several programs, the Chinese government has recognized innovation and knowledge transfer to be the engine of economic development [23,24]. In 2012, China launched its Innovation-driven Development Strategy. In 2015, the Law of Promoting the Application of Scientific and Technological Achievements, originally launched in 1996, was revised to encourage research and development (R&D) organizations and universities to transfer technologies to enterprises [16]. The fast growth in the number of Chinese patent applications translates to different knowledge transfer patterns within China [25], and due to the implementation of the filing system, patent licensing is regarded as the main measurable form of technology transfer in China [21]. Chinese universities represent one of the world's largest groups in academic research, and technology transfer is one of this system's central roles. Academic interest in Chinese university technology transfer in both the West and China has also increased in parallel [26].

### 2.2. Recent Studies about SNA on Technology Transfer

Social networks are increasingly considered to be influential in explaining the process of knowledge transfer [16]. The network positions of nodes allow us to identify the more prominent, i.e., more important, firms within the university–firm knowledge transfer network. Furthermore, recent developments in the field of SNA have resulted in software tools for visualization, such as *Gephi* (https://gephi.org/, accessed on 2 August 2022.) and *Pajek* (See http://mrvar.fdv.uni-lj.si/pajek/, accessed on 2 August 2022.), which can facilitate the analysis and interpretation of patent statistics, e.g., patent applications, patent citations, joint patent applications [27–29], and patent license transfers.

In studies about China, previous studies have researched the role of universities in the university knowledge transfer network, e.g., ref. [16] indicated that key universities have high centrality scores within a network, thus allowing them to gain control of and easier access to knowledge. On the other hand, although Chinese overall innovation capability has improved, innovation capabilities, R&D resources, and technology transfer performance vary across China's eastern, central, and western areas due to uneven economic development. The authors of ref. [30] reported that the uneven distribution of innovative activities among regions has given rise to a phenomenon of technology transfer from R&D resource-intensive areas to industrialized areas. For example, a large number of technologies that were invented in Beijing, a recognized center for technological innovation and diffusion with powerful scientific, economic, and political strength, have been transferred to other industrialized provinces, such as Zhejiang, Jiangsu, and Guangdong [31]. The authors of ref. [21] investigated licensed patents at the provincial level and found that patent licensing networks present clear small-world phenomena. For studies about other countries, ref. [32] reported an analysis on two-mode networks emerging from the relations between national and manufacturing firms that may find the destination of their products, general activities in Colombia, and proposed a methodology to investigate the structure of innovation from a survey on innovation and technological development. The authors of ref. [33] analyzed the determinants of countries' embeddedness in the global photovoltaics knowledge network and found that the number of networks of international research collaboration are constantly growing, and while European countries collaborate quite frequently with international partners, Asian countries conduct most of their research domestically. The authors of ref. [34] developed and exploited a novel database on patent co-authorship to investigate the effects of collaboration networks on innovation. They constructed regional collaboration networks for moving five-year windows in all 337 US Metropolitan Statistical Areas. Their discovery was that collaboration networks are growing, along with the quantitative and qualitative evidence that spillovers occur quite easily along current and historical collaborative ties, and they implied that managers must pay increasing attention to the incentives, socialization, and collaborative opportunities of their primary inventors. The authors of ref. [35] proposed a quantitative analysis of the social distance between Open Science and Proprietary Technology by investing Italian scientific and academic collaboration networks. They showed that academic inventors to be more central and better connected than non-academic ones, and they play key roles in connecting individuals and network components.

Furthermore, many large real-world networks actually have a two-mode nature: their nodes may be separated into two classes, the links being between nodes of different classes only [36]. When analyzing a bipartite network, the common method is to change the bipartite network into a unipartite network, which can then be analyzed with standard techniques [32,37]. However, unipartite projections often destroy important structural information [38]; for example, they only give the centrality estimate for either mode of the network and fail to consider the weight of edges connecting both modes. In order to solve this problem, bipartite ranking algorithms such as CoHITS, BGRM, and BiRank have been developed. The authors of ref. [39] measured centrality with recently-developed bipartite methods, and found that BiRank and CoHITS provide significantly more robust measures of prescription drug-seeking and better predictors of subsequent opioid overdose

than traditional centrality estimates. The authors of ref. [40] proposed an approach called VenueRank, which consists of employing bipartite graph ranking algorithms; they also use well-established algorithms, such as BiRank, to help discover important venues in artist-venue graphs mined from Facebook. The authors of ref. [41] proposed a solution to the time-bias problem of ranking nodes in bipartite networks, using the structural property of the network and the centrality of the nodes.

Although there are sufficient studies about defining central positions in social network theoretically, there are still few studies that have focused on how to apply them to the real world, especially to the case of Chinese industrial firms.

## 3. Materials and Methods

### 3.1. Schematic of Research Procedure

Figure 1 shows the schematic of our research procedure. Overall, the knowledge and innovation generated by researchers at universities is transferred to industries through patent licensing, bringing innovative output and commercialization. In order to investigate the development of Chinese university–industry technology transfer and whether this kind of collaboration may affect a firm's innovation output, first, we collect contracts made between Chinese manufacturing firms and universities. Next, we make a bipartite network base on these contracts. Then, we allocate a centrality estimate to every firm using the SNA method. In this step, we also give a visualization of the dynamics of the network and centrality estimates. After that, we collect other factors about the number of employees and the history of firms. Finally, we conduct empirical analysis on how centrality estimates and these factors affect firms' innovation ability.

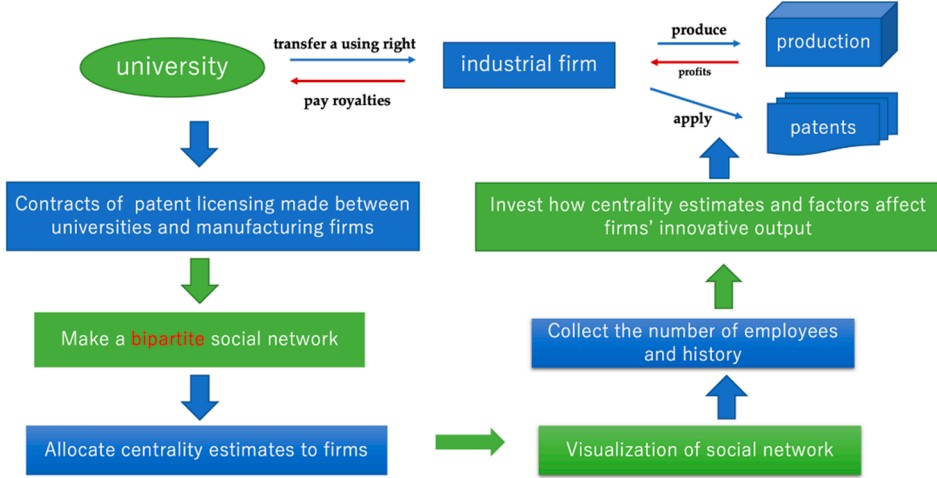

**Figure 1.** Schematic of research procedure.

### 3.2. License Transfer Data Collection

We collected information about approximately 6000 license contracts made between more than 4000 Chinese manufacturing firms and 300 Chinese universities during the period between 2009 and 2014. The data for license contracts consist of the names of the licensor and licensee, the year the contract was made, the type of license, and the patent number for the license contract. The raw data were all collected from the online database of the Chinese State Intellectual Property Office (SIPO). We made a bipartite network of license transfers from universities to firms, and then extracted 876 pairs of license contracts that form a complex network. Finally, we constructed a database for empirical analysis, which includes license contracts between 95 Chinese universities and 755 Chinese industrial firms.

### 3.3. Data Collection from Firms

Technological innovation is regarded as a critical driving element that enhances an enterprise's sustainable competitive advantage, productivity, and growth. In this paper, we

use patent applications as indicators of the innovation of a firm and investigate how their centrality score may influence their innovation output. We also use firm-level data such as the number of laborers and the firm's history. We obtained data for various firms from the SIPO.

We collected information on the number of patent applications by the abovementioned firms associated with university license contracts from the SIPO for the period between 2009 and 2021. We utilized patent applications as an indicator of innovative output and examined the number of employees at these firms and the firms' year of establishment. We also collected information on the locations of the universities and firms (province and city name), as well as their longitudes and latitudes using Google Maps.

Figure 2 shows sample data for the patent applications and license contracts made with universities per year for ten randomly selected firms from the dataset. From this sample, it can be observed that the number of most firms' patent applications increased with time also after making license transfer contracts with universities.

### 3.4. Unipartite and Bipartite Networks

A social network is a representation of a system in which nodes are connected by ties, and it is also a set of specifications and methods for characterizing the structure and attributes of the relationships formed by social actors [42]. An advantage of this type of network is that it can quantify the relationships between actors and their connections rather than relying on traditional attribute data. It can also establish a relational model between objects, providing descriptions of the network characteristics and interactions between actors. Network actors can be individuals, institutions, regions, or even countries [21]. In our study, the nodes represent the universities and firms that have license contracts.

While most networks are defined as unipartite networks, which have only one set of similar nodes, several networks are in fact bipartite networks [19,36]. Bipartite networks are characterized by having two different sets of nodes, with ties existing only between nodes belonging to different sets.

Figure 3 shows a schematic diagram of our bipartite network model for university–firm technology transfer (no real data are shown). In our research, there are two sets of nodes, one set representing universities, and the other, firms. Patent licenses indicate the source and destination of the flow of knowledge and technology [21]. Once a university (round node) transforms a use right for a patent to a firm (square node), a line will connect the two nodes across the two sets. Additionally, the width of edges reflects how many contracts are made between one university and its counterpart firm; the more contracts made, the wider the edges are.

### 3.5. Centrality Indices in Bipartite Social Network

In this section, we focus on the centrality score for firms in the network. Bipartite ranking algorithms such as CoHITS, BGRM, and BiRank have been developed; these algorithms operate similarly to eigenvector-based methods in that they iteratively update node centrality estimates based on each node's connectivity and walk distance to other prominent nodes in the social network [39]. The author of ref. [18] proposed the Hyperlink-Induced Topic Search (HITS), a link analysis algorithm that evaluates the importance of web pages. The main idea of the HITS algorithm is that the number of web pages referenced and the number of other sites that link to it are used to calculate a page's authority and hub value, respectively. This method has been used to rank the importance of journals and websites. However, when considering a bipartite network, the HITS algorithm only considers the content and link information from one side. The authors of ref. [43] proposed a novel and generalized CoHITS algorithm that incorporates a bipartite graph with content information from both sides. Generally, CoHITS estimates the bipartite ranks (centrality estimates) of nodes from an edge list or adjacency matrix. Then, the bipartite graph reinforcement model (BGRM) was developed for automating web image annotation [44]. BGRM differs from CoHITS centrally in its use of a symmetrical weighting scheme, whereby all edges are

normalized by both vertexes. BiRank is the newest of the bipartite centrality algorithms and was developed to improve upon the theoretical advantages of BGRM's symmetric normalization scheme [45]. The BiRank algorithm generates the ranking values for nodes from both sides simultaneously and takes into account the full network topology without any information loss [46]. It is worth mentioning that assigning different edge weights to the network can significantly affect score estimates.

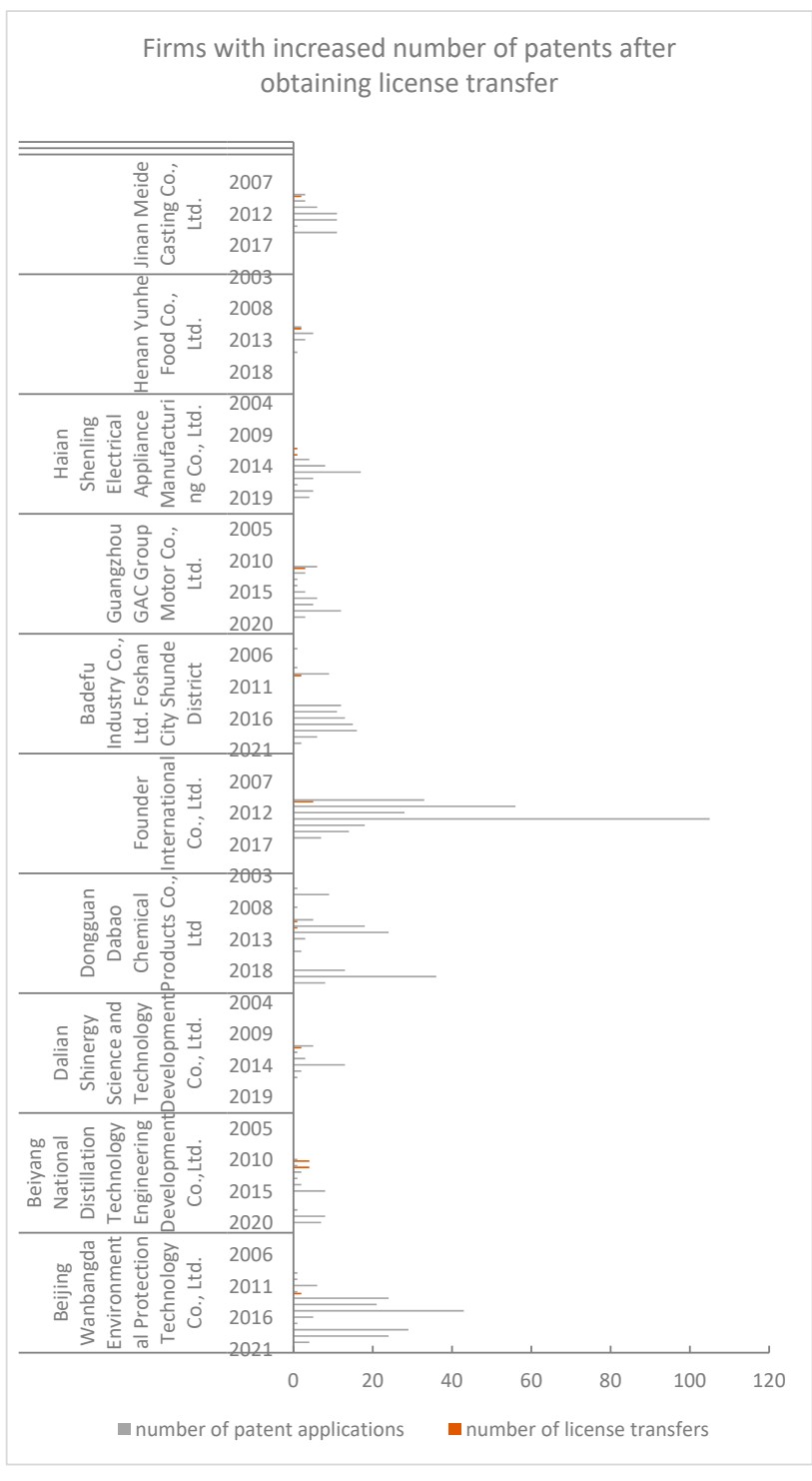

**Figure 2.** Sample data for the number of patent applications and license transfer contracts firms made with universities per year.

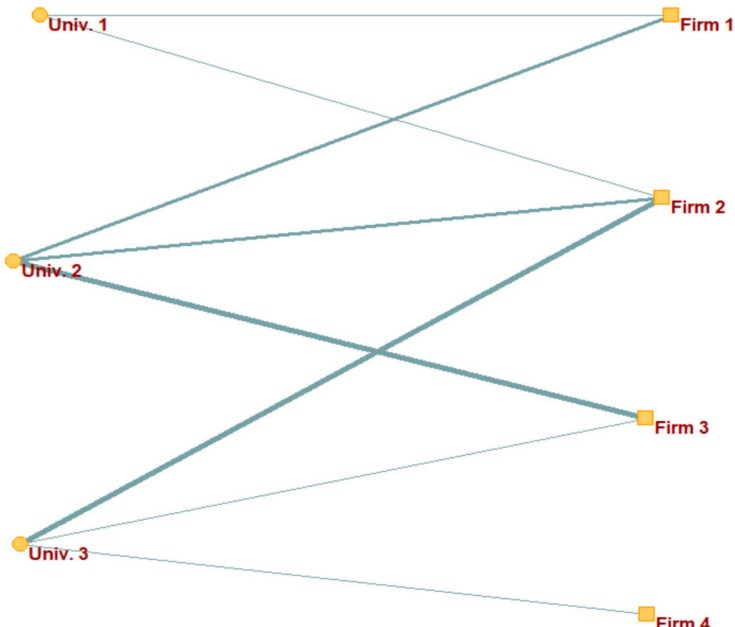

**Figure 3.** Schematic of bipartite network model for university–firm technology transfer.

In this study, we utilized the number of contracts between one university and its counterpart firm as the weight of the edge. In order to incorporate edge weights into score estimations, we chose the BGRM and BiRank models to test our hypothesis. We acquired the BGRM and BiRank estimates through the R package "birankr" (See https://cran.r-project.org/web/packages/birankr/birankr.pdf, accessed on 2 August 2022).

*3.6. Empirical Analysis Method*

Patent data have been used as a source of information both on the extent of invention and on the value of the protection generated by patent laws [47]. In the literature, patent application counts are usually taken to represent a measure of innovative output. Thus, we formulate a hypothesis that a firm with a higher centrality index will have greater innovative output. Some other factors such as the number of employees and the history of the firm are also thought to affect a firm's innovative ability.

Considering the number of patents associated with a firm to constitute a frequency, we assume that the $i$th firm's patent count is drawn from the Poisson distribution, that is,

$$N_i \sim \text{Possion}(\lambda_i).$$

This assumption ensures that any patent is a nonnegative integer, and the expectation of $N_i$ is $\lambda_i$. We can then formulate a Poisson regression model as follows:

$$\log\left(\lambda_i^k\right) = \beta_0^k + \beta_1^k \cdot \text{history}_i + \beta_2^k \cdot \text{labor}_i + \beta_2^k \cdot \text{centrality}_i^k$$

for $i = 1, \ldots, 675$ and $k \in \{\text{BGRM, BiRank}\}$. In the above model, the explanatory variable $\text{history}_i$ represents the years elapsed since firm $i$ was established. The explanatory variable $\text{labor}_i$ is the number of employees at firm $i$, and $\text{centrality}_i^k$ represents the BGRM and BiRank centrality index calculated for firm $i$ in our bipartite network. The intercept $\beta_0$ and coefficients $\beta_1$, $\beta_2$ are estimated relative to the BGRM and BiRank centrality.

Note that the probability of $N_i$ conditional on $\lambda_i$ is given as

$$\Pr(N_i|\lambda_i) = \frac{\lambda_i^{N_i} \cdot \exp(-\lambda_i)}{N_i!},$$

where $N_i$ is the number of a firm's patent applications recorded by the SIPO during the period between 2009 and 2021.

## 4. Results

### 4.1. Overview of Full Network Representing Chinese University-Firm Knowledge Transfer

Figure 4 shows the network of all license contracts made between more than 4000 Chinese manufacturing firms and 300 Chinese universities. We can observe that this network is not fully connected, and more specifically, that one part is strongly connected while many small parts are scattered peripherally. As centrality is not meaningful in an extremely simple structure such as a star structure or a linear mode sub-network, we used only the strongly connected part for the next part of the analysis involving allocating centrality estimates to every node.

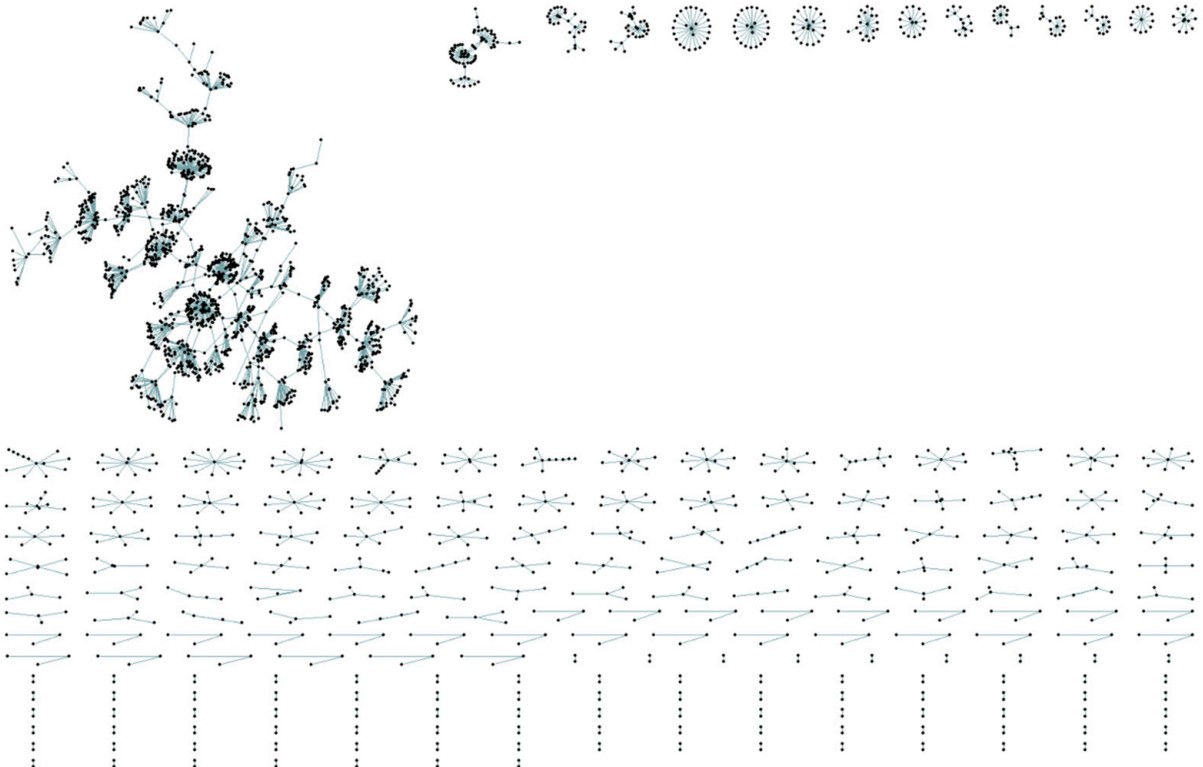

**Figure 4.** Visualization of full network representing Chinese university–firm knowledge transfer.

### 4.2. Visualization of Dynamics of Chinese University–Firm Knowledge Transfer Network

Using the geographical information collected from Google Maps, we were able to trace knowledge flows with the help of the SNA visualization tool, *Gephi*. In the present study, we visualized the university knowledge transfer network and observed its year-to-year evolution by setting the year a contract was made as the time truncation. We then took snapshots of the network for license transfers in 2009, 2011, and 2013 to visualize and capture the dynamics of Chinese university–firm technology transfer networks.

Figures 5–7 show snapshots of the node connections representing Chinese university–firm license transfers in 2009, 2011, and 2013 respectively. The nodes are mapped according to the geographical locations associated with the universities and firms. We can observe some features from these visualizations: the number of edges (contracts) from universities to firms rapidly increased from 2009 to 2011, and conversely decreased from 2011 to 2013. The license transfer activities among regions are also quite uneven: nearly all universities and firms have patent license transfer contracts located in China's southeastern, economically developed areas, with the most license transfers located in the Yangtze River economic

zone, followed by the Pearl River Delta economic zone. In northern China, most nodes are clustered around Beijing and Tianjin. Additionally, rarely do we observe long edges that cross the entire map, that is, most universities and firms locate in the same province, or in surrounding provinces. From these features, we can conclude that, in agreement with previous studies [16], innovation capabilities, R&D resources, and technology transfer performance all vary across China, and patent licensing networks present clear small-world phenomena.

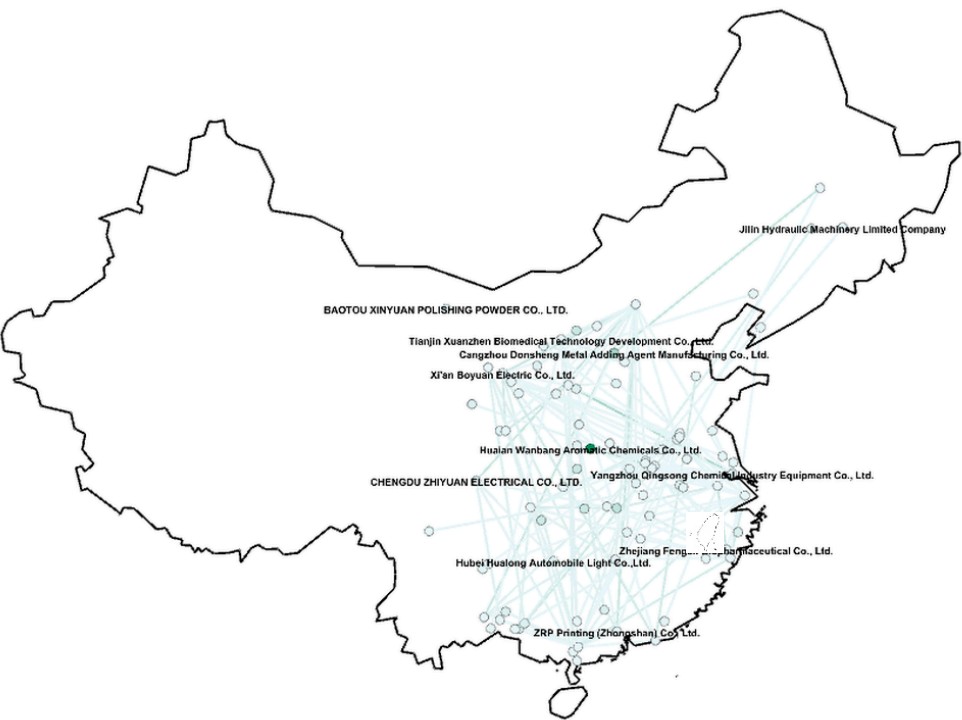

**Figure 5.** Geographically mapped Chinese university–firm knowledge transfer network for 2009.

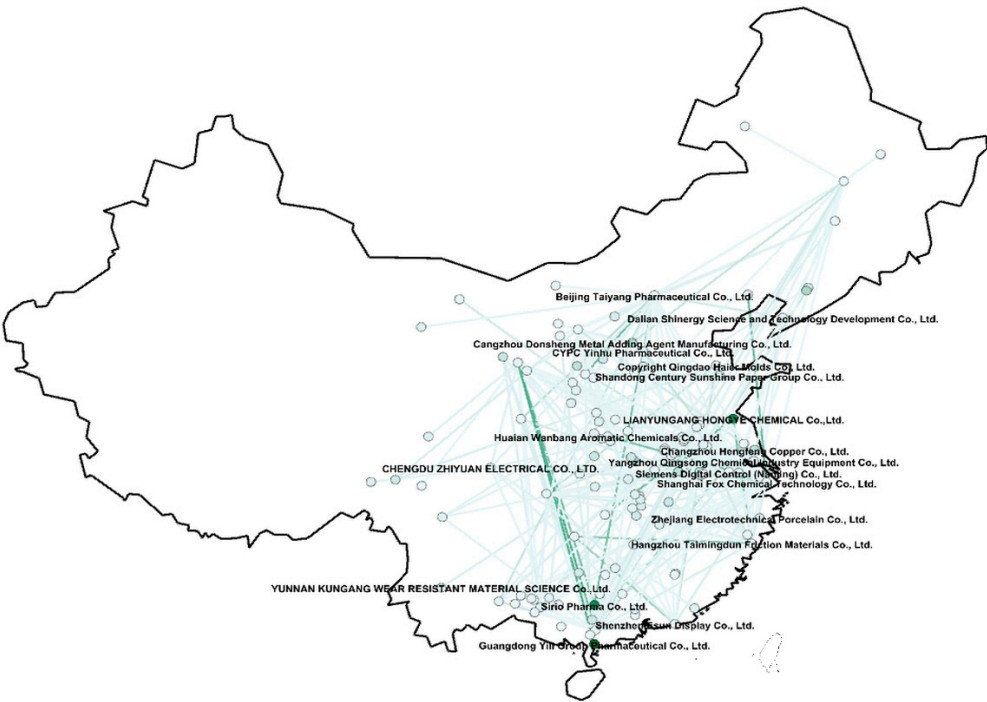

**Figure 6.** Geographically mapped Chinese university–firm knowledge transfer network for 2011.

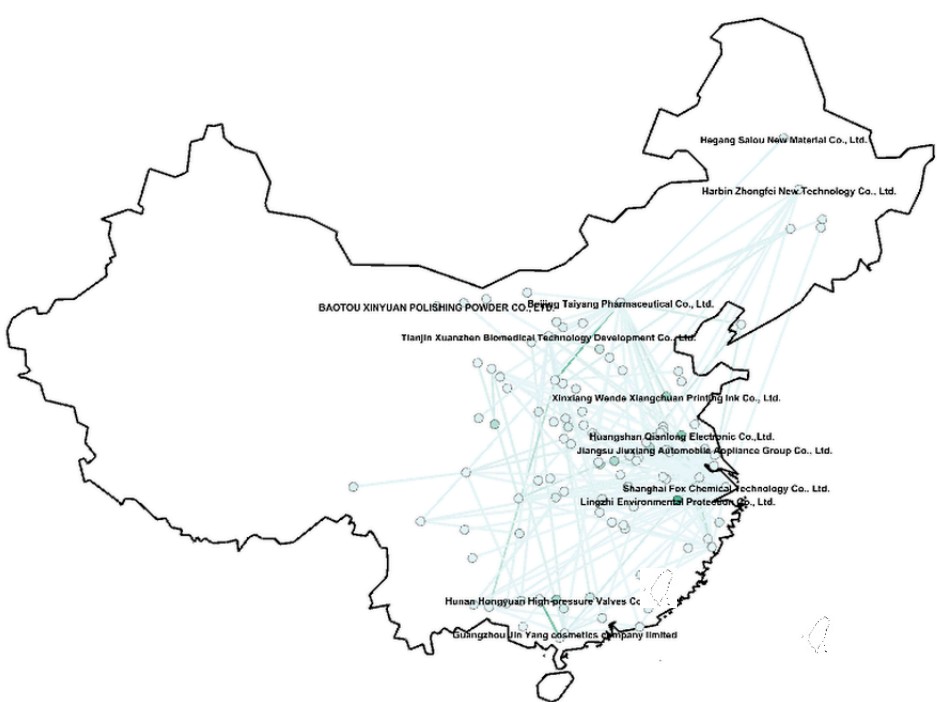

**Figure 7.** Geographically mapped Chinese university–firm knowledge transfer network for 2013.

*4.3. Visualization of Geographically Mapped Centrality Scores*

Figure 8 shows the association between the BGRM centrality of nodes and their geographical location. We calculated BGRM and BiRank centrality based on the whole sample from 2009 to 2014, so that it also shows the overview of the full network. The size and coloration of the nodes reflects their centrality score, with larger and darker nodes corresponding to firms that have high centrality, and small, light-colored nodes indicating low centrality. We also investigated the relation between the firms' centrality and their geographic features. We can observe that almost all universities and firms with high centrality locate in China's southeastern, economically developed areas, with the most being in the Yangtze River economic zone and the Pearl River Delta economic zone. In northern China, most license transfer activities are located around Beijing and Tianjin. Furthermore, as the width of the edges reflects how many contracts are made between one university and its counterpart firms, we can observe a positive correlation between the size of nodes and the width of the edges connected with it; that is to say, the more license contracts made by a firm, the higher its centrality score.

*4.4. Visualization and Summarization of Centrality Score*

We then examined the centrality of the network with separate nodes for the two different measures of centrality. Figure 9 shows the BGRM centrality in shades of green, while Figure 10 shows the BiRank centrality in red. The size and coloration of the nodes reflects their centrality score, with larger and darker nodes corresponding to firms that have high centrality, and small, light-colored nodes indicating low centrality. Some of the observed features are as follows: firms having high BGRM centrality have a tendency to also have high BiRank centrality, e.g., Guangdong Yili Group Pharmaceutical Co., Ltd. (Chiyoda City, Tokyo) and Shanghai Fox Chemical Technology Co., Ltd. (Shanghai, China). However, a few firms are observed to have high BiRank centrality but low BGRM centrality, e.g., Tianjin Xuanzhen Biomedical Technology Development Co., Ltd. (Tianjin, China). There are also firms with high BGRM centrality but low BiRank centrality, e.g., Beijing Taiyang Pharmaceutical Co., Ltd. (Beijing, China). In the next section, we discuss how BiRank and BGRM centrality affect a firm's innovative output.

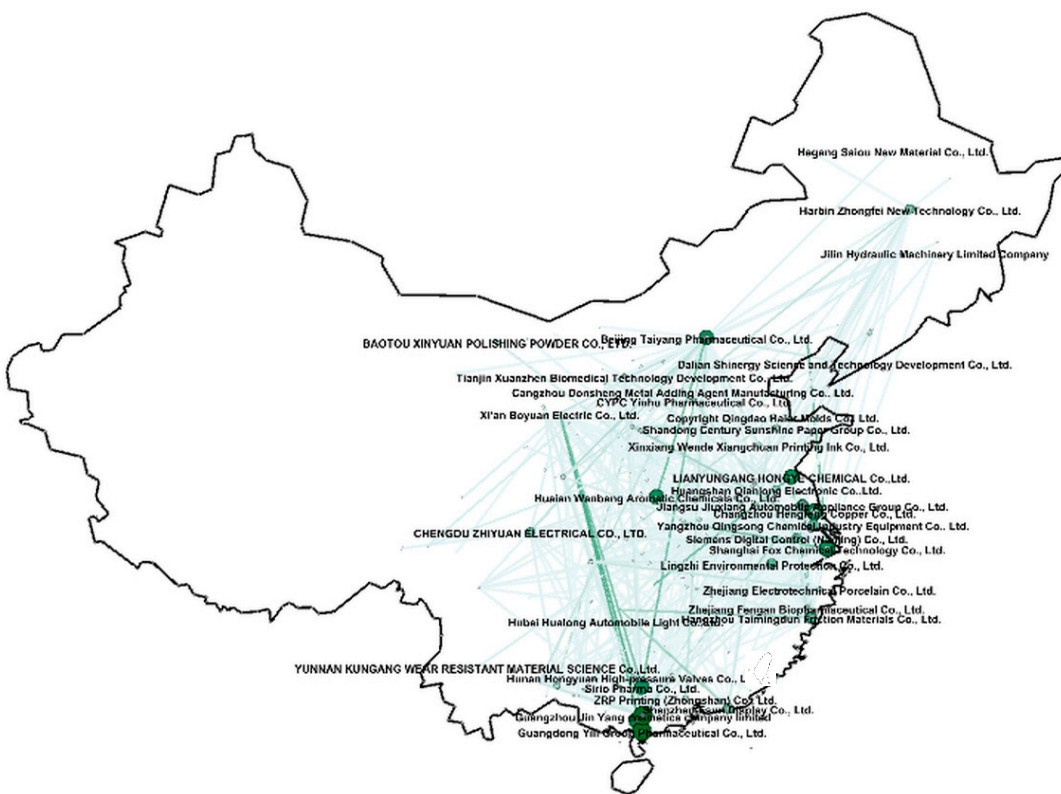

**Figure 8.** Geographically mapped BGRM centrality of firms.

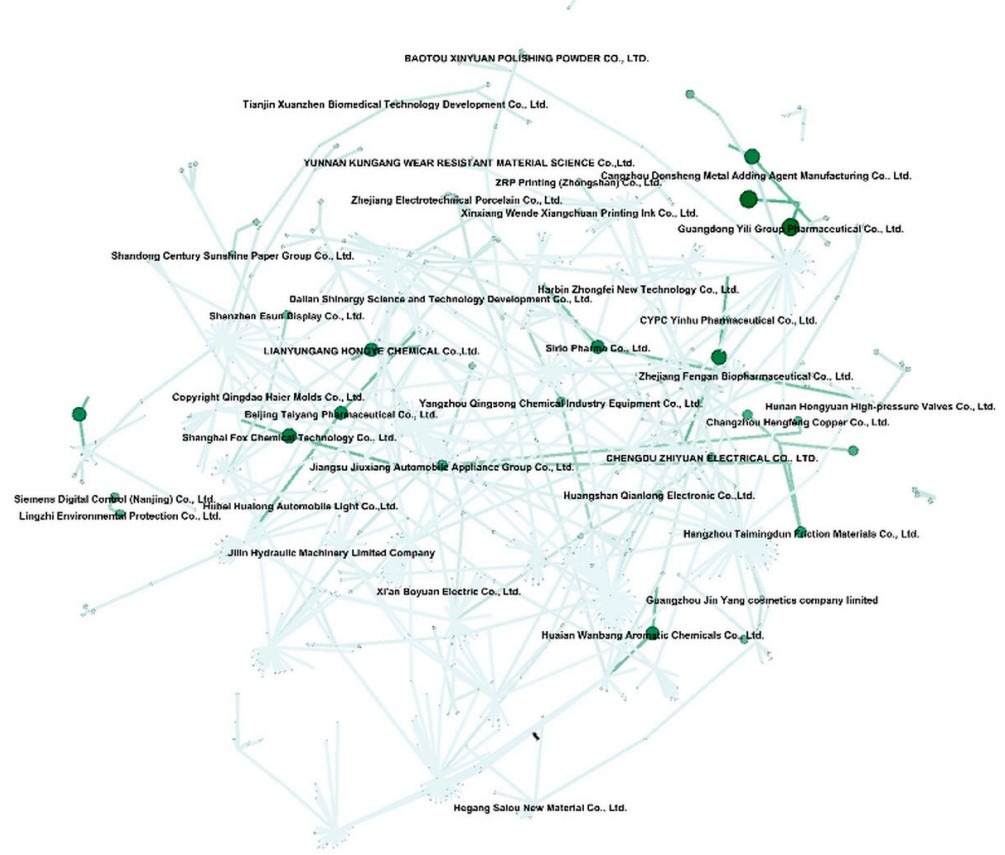

**Figure 9.** Diagram of network with BGRM centrality.

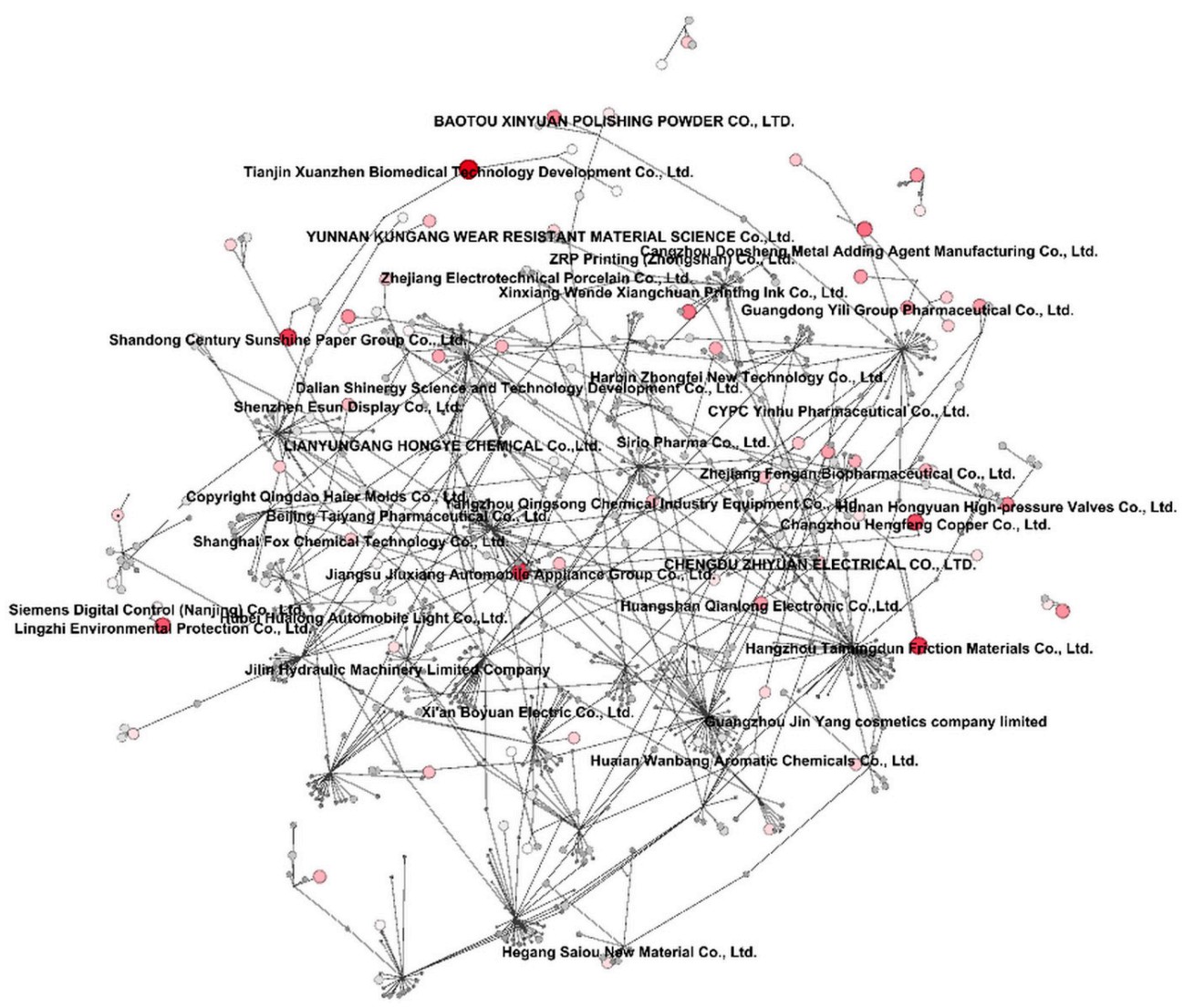

**Figure 10.** Diagram of network with BiRank centrality.

The top firms with the highest BGRM and BiRank centrality scores are listed in Table 1 in order of descending BGRM centrality. We can observe that there is indeed a relatively strong positive correlation between the BGRM centrality and BiRank centrality estimates. We will verify this point in next paragraph.

**Table 1.** Firms with highest BGRM and BiRank centrality.

| Name of Firm | BGRM Centrality | BiRank Centrality |
|---|---|---|
| Guangdong Yili Group Pharmaceutical Co., Ltd. | 0.0013824 | 0.004137 |
| Changzhou Fanqun Drying Equipment Co., Ltd. | 0.0012163 | 0.0041057 |
| Shanghai Fox Chemical Technology Co., Ltd. | 0.0011935 | 0.0037608 |
| Shanghai Taiho Paint Products Co., Ltd. | 0.0011187 | 0.0036851 |
| Lianyungang Hongye Chemical Co., Ltd. | 0.0010962 | 0.0032432 |
| Huaian Wanbang Aromatic Chemicals Co., Ltd. | 0.0010925 | 0.0032684 |
| Sirio Pharma Co., Ltd. | 0.0010862 | 0.0030402 |
| Beijing Taiyang Pharmaceutical Co., Ltd. | 0.0010843 | 0.0031152 |
| Hangzhou Taimingdun Friction Materials Co., Ltd. | 0.0009244 | 0.0050411 |

**Table 1.** *Cont.*

| Name of Firm | BGRM Centrality | BiRank Centrality |
|---|---|---|
| Jiangsu Jiuxiang Automobile Appliance Group Co., Ltd. | 0.0008878 | 0.0049912 |
| Jiangmen Kingbord Laminates Holdings Ltd. | 0.0008149 | 0.003535 |
| Yangzhou Qingsong Chemical Industry Equipment Co., Ltd. | 0.000785 | 0.0038608 |
| Lingzhi Environmental Protection Co., Ltd. | 0.0007702 | 0.0047376 |
| Siemens Digital Control (Nanjing) Co., Ltd. | 0.0007702 | 0.0034082 |
| Changzhou Hengfeng Copper Co., Ltd. | 0.0007416 | 0.0048171 |
| Shenzhen Esun Display Co., Ltd. | 0.000724 | 0.0038745 |
| Chengdu Zhiyuan Electrical Co., Ltd. | 0.0007233 | 0.0037257 |
| Jiangsu Dazu Yueming Laser Technology Co., Ltd. | 0.0007232 | 0.0037223 |
| Shandong Century Sunshine Paper Group Co., Ltd. | 0.0005875 | 0.0050004 |
| Huangshan Qianlong Electronic Co., Ltd. | 0.000569 | 0.0041424 |
| Baotou Xinyuan Polishing Powder Co., Ltd. | 0.0005021 | 0.0034874 |
| Zhejiang Fengan Biopharmaceutical Co., Ltd. | 0.0005018 | 0.0031829 |
| CYPC Yinhu Pharmaceutical Co., Ltd. | 0.0005018 | 0.0023089 |
| ZRP Printing (Zhongshan) Co., Ltd. | 0.0004977 | 0.0030937 |
| Xinxiang Wende Xiangchuan Printing Ink Co., Ltd. | 0.0004977 | 0.0030937 |
| Zhejiang Electrotechnical Porcelain Co., Ltd. | 0.0004928 | 0.0031293 |
| Copyright Qingdao Haier Molds Co., Ltd. | 0.0004869 | 0.0029626 |
| Tianjin Xuanzhen Biomedical Technology Development Co., Ltd. | 0.0004801 | 0.0055489 |

*4.5. Results of Empirical Analysis*

4.5.1. Correlation Test

The correlations between the explanatory variables and the BGRM and BiRank centrality indices are given in Table 2.

**Table 2.** Results of correlation test.

| | Labor | History | BGRM | BiRank |
|---|---|---|---|---|
| Labor | 1 | 0.27 | 0.06 | −0.11 |
| History | | 1 | 0.02 | −0.07 |
| BGRM | | | 1 | 0.63 |
| BiRank | | | | 1 |

It can be observed that there is a weak correlation between the explanatory variables, labor and history, and the BGRM and BiRank centrality indices. Furthermore, the BiRank index seems to correlate with the explanatory variables in a different manner compared to BGRM. Considering that both BiRank and BGRM are commonly used centrality indices, this observation indicates that it would be meaningful to apply the generalized linear regression framework to investigate whether the choice of the centrality index affects the estimation results.

4.5.2. Empirical Analysis Results for BGRM and BiRank Models

Due to there being missing values for the labor variable, we applied the method of multiple imputation and created ten different datasets. The multiple imputation is a flexible method for handling the missing data and can be implemented by using the R package "mice". Figure 11 provides the stripplots of observed and imputed data. Stripplot number zero on the horizontal axis represents the original dataset with missing values. Stripplots numbers one through ten on the horizontal axis are the ten different datasets created using the method of multiple imputation, where the red points are the imputed data for the labor variable. Figure 12 further depicts the densities of the observed and imputed data for the labor variable. The density plots are combined into a single panel and show that the ten sets of imputed data (i.e., red lines) have a very similar distribution to the observed data for the labor variable (i.e., the blue line). Therefore, the method of multiple imputation can be considered as a useful solution to the missing data in this example and the ten different datasets can be used as alternatives of the original dataset.

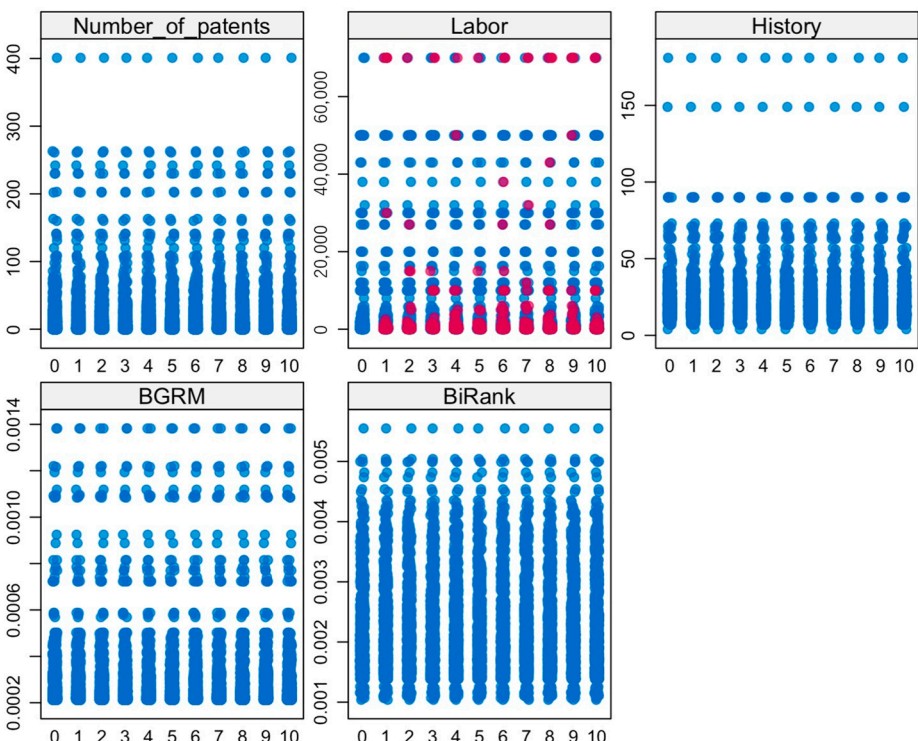

**Figure 11.** Stripplots of the observed and imputed data (blue is observed and red is imputed).

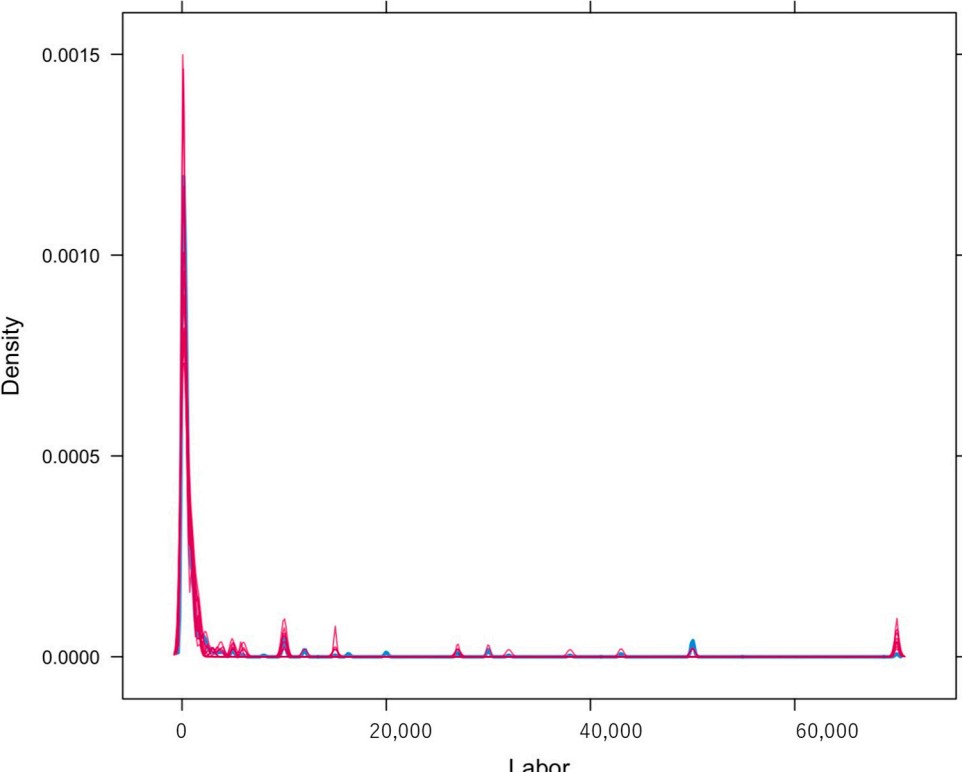

**Figure 12.** Density plots of the observed and imputed data for the labor variable. (Blue is observed and red is imputed).

The estimated results of the BGRM and BiRank models using datasets (1) through (10) are given in Tables 3 and 4. Since the ten datasets created using the method of multiple imputation share a very similar distribution to each other and to the original dataset,

the estimated results are extremely similar. This is a good observation, because if the estimates differ significantly among the different data sets, the missing data issue needs to be reconsidered.

**Table 3.** Empirical analysis results for BGRM centrality using imputed datasets.

| | Dependent Variable | | | | | | | | | |
|---|---|---|---|---|---|---|---|---|---|---|
| | Number of Patents | | | | | | | | | |
| | (1) | (2) | (3) | (4) | (5) | (6) | (7) | (8) | (9) | (10) |
| Labor | −0.00001 *** | −0.00001 *** | −0.00001 *** | −0.00001 *** | −0.00001 *** | −0.00001 *** | −0.00001 *** | −0.00001 *** | −0.00001 *** | −0.00001 *** |
| | (0.0000) | (0.0000) | (0.0000) | (0.0000) | (0.0000) | (0.0000) | (0.0000) | (0.0000) | (0.0000) | (0.0000) |
| History | 0.0120 *** | 0.0120 *** | 0.0120 *** | 0.0120 *** | 0.0120 *** | 0.0120 *** | 0.0120 *** | 0.0120 *** | 0.0120 *** | 0.0120 *** |
| | (0.0004) | (0.0005) | (0.0006) | (0.0007) | (0.0008) | (0.0009) | (0.0010) | (0.0011) | (0.0012) | (0.0013) |
| BGRM | 128.8060 ** | 128.8060 ** | 128.8060 ** | 128.8060 ** | 128.8060 ** | 128.8060 ** | 128.8060 ** | 128.8060 ** | 128.8060 ** | 128.8060 ** |
| | (62.4590) | (62.4600) | (62.4610) | (62.4620) | (62.4630) | (62.4640) | (62.4650) | (62.4660) | (62.4670) | (62.4680) |
| Constant | 2.6020 *** | 2.6020 *** | 2.6020 *** | 2.6020 *** | 2.6020 *** | 2.6020 *** | 2.6020 *** | 2.6020 *** | 2.6020 *** | 2.6020 *** |
| | (0.0230) | (0.0240) | (0.0250) | (0.0260) | (0.0270) | (0.0280) | (0.0290) | (0.0300) | (0.0310) | (0.0320) |
| Observations | 675 | 675 | 675 | 675 | 675 | 675 | 675 | 675 | 675 | 675 |
| Log-likelihood | −11,954.3000 | −11,954.3000 | −11,954.3000 | −11,954.3000 | −11,954.3000 | −11,954.3000 | −11,954.3000 | −11,954.3000 | −11,954.3000 | −11,954.3000 |
| Akaike Inf. Crit. | 23,916.5900 | 23,916.5900 | 23,916.5900 | 23,916.5900 | 23,916.5900 | 23,916.5900 | 23,916.5900 | 23,916.5900 | 23,916.5900 | 23,916.5900 |

Note: ** $p < 0.05$; *** $p < 0.01$.

**Table 4.** Empirical analysis results for BiRank centrality using imputed datasets.

| | Dependent Variable | | | | | | | | | |
|---|---|---|---|---|---|---|---|---|---|---|
| | Number of Patents | | | | | | | | | |
| | (1) | (2) | (3) | (4) | (5) | (6) | (7) | (8) | (9) | (10) |
| Labor | −0.00001 *** | −0.00001 *** | −0.00001 *** | −0.00001 *** | −0.00001 *** | −0.00001 *** | −0.00001 *** | −0.00001 *** | −0.00001 *** | −0.00001 *** |
| | (0.0000) | (0.0000) | (0.0000) | (0.0000) | (0.0000) | (0.0000) | (0.0000) | (0.0000) | (0.0000) | (0.0000) |
| History | 0.0120 *** | 0.0120 *** | 0.0120 *** | 0.0120 *** | 0.0120 *** | 0.0120 *** | 0.0120 *** | 0.0120 *** | 0.0120 *** | 0.0120 *** |
| | (0.0004) | (0.0004) | (0.0004) | (0.0004) | (0.0004) | (0.0004) | (0.0004) | (0.0004) | (0.0004) | (0.0004) |
| BiRank | 79.3500 *** | 79.3500 *** | 79.3500 *** | 79.3500 *** | 79.3500 *** | 79.3500 *** | 79.3500 *** | 79.3500 *** | 79.3500 *** | 79.3500 *** |
| | (12.3950) | (12.3950) | (12.3950) | (12.3950) | (12.3950) | (12.3950) | (12.3950) | (12.3950) | (12.3950) | (12.3950) |
| Constant | 2.4660 *** | 2.4660 *** | 2.4660 *** | 2.4660 *** | 2.4660 *** | 2.4660 *** | 2.4660 *** | 2.4660 *** | 2.4660 *** | 2.4660 *** |
| | (0.0310) | (0.0310) | (0.0310) | (0.0310) | (0.0310) | (0.0310) | (0.0310) | (0.0310) | (0.0310) | (0.0310) |
| Observations | 675 | 675 | 675 | 675 | 675 | 675 | 675 | 675 | 675 | 675 |
| Log-likelihood | −11,936.3900 | −11,936.3900 | −11,936.3900 | −11,936.3900 | −11,936.3900 | −11,936.3900 | −11,936.3900 | −11,936.3900 | −11,936.3900 | −11,936.3900 |
| Akaike Inf. Crit. | 23,880.7700 | 23,880.7700 | 23,880.7700 | 23,880.7700 | 23,880.7700 | 23,880.7700 | 23,880.7700 | 23,880.7700 | 23,880.7700 | 23,880.7700 |

Note: *** $p < 0.01$.

From the results listed in Tables 3 and 4, we can observe that the coefficients are positive and significant both for the BGRM and BiRank. This implies that firms with high centrality usually have greater innovative output. The coefficient is also positive and significant for history, which means firms with a long history will have greater innovative output. However, the coefficient is negative and significant for the number of employees, which may be contrary to expectations.

## 5. Discussion

Overall, our research answers some important methodological questions posed by previous studies, such as how to investigate every player's centrality in a bipartite network without loss of information, and how to capture network dynamics while considering geographical features. In comparison with former research on Europe and the US, our findings are in accord with [33,34] in that the technology transfer network is growing, and present clear small-world phenomena according to the geographical features or economic developing situation. However, due to the fact that there is limited research and it is difficult to collect each player's exact location information, the coast and capital-oriented feature of the Chinese university–industry technology transfer network cannot be compared with existing studies. Furthermore, the empirical analysis results reveal that firms with high centralities have greater innovative output, hence we recommend that, for industrial firms

that lack independent innovation ability, it is a good idea to seek collaborations with universities to improve this situation.

Our research also has some limitations. For example, we were not able to obtain data on the license contracts made between Chinese universities and Chinese manufacturing firms from the online database of SIPO for dates after 2014, so the time period analyzed was short. We also were not able to collect characteristics for universities, but the BGRM and BiRank algorithms still allocated centrality estimates to them. We will carry out further research to overcome these issues in the future.

## 6. Conclusions

We employed a bipartite SNA technique to empirically analyze the characteristics of firms in a university–industry knowledge transfer network. We calculated centrality scores in the bipartite network, emphasizing visual exploration, and utilized patent applications as output indicators for R&D investment. We combined these various characteristics to investigate their effects on firms' innovative output. Our main results are summarized as follows.

1.  When analyzing a bipartite network, a common analysis method involves changing the bipartite network into a unipartite network, which can then be analyzed with standard techniques. However, unipartite projections often destroy important structural information. Our research serves to close this gap by giving each node in a bipartite network a centrality estimate, while still considering the edge weight, e.g., the number of license contracts between paired universities and firms.

2.  Previously, few studies have captured how knowledge transfer networks evolve over time or combined the time series with geographical features, e.g., visualized knowledge transfer network dynamics on a map. We created visualizations of the university knowledge transfer network and observed its year-by-year evolution by setting the licensing year as the time truncation using the SNA tool *Gephi*. We used snapshots representing license transfers in 2009, 2011, and 2013 to visualize and capture the dynamics of Chinese university–firm technology transfer. Our visualization results showed that nearly all universities and firms that have patent license transfer contracts are in China's southeastern, economically developed areas, with the most license transfers in the Yangtze River economic zone and the Pearl River Delta economic zone. In northern China, most universities and firms are clustered around Beijing and Tianjin. Furthermore, in accord with previous research, we observed that innovation capabilities, R&D resources, and technology transfer performance varied across China, and that patent licensing networks present clear small-world phenomena.

3.  We highlighted BGRM and BiRank centrality in a bipartite network and investigated the relationship between them. We found that firms having high BGRM centrality most often also have high BiRank centrality, and there is a relatively strong positive correlation between the BGRM centrality and BiRank centrality estimates.

4.  Our empirical analysis results revealed that firms with high BGRM or BiRank centralities have greater innovative output. Furthermore, firms with a long history and fewer employees also have greater innovative output.

**Author Contributions:** Conceptualization, J.J.; methodology, J.J. and Y.Z.; software, J.J. and Y.Z.; data curation, J.F.; writing—original draft preparation, J.J.; writing—review and editing, Y.Z. and J.F. All authors have read and agreed to the published version of the manuscript.

**Funding:** This research received no external funding.

**Institutional Review Board Statement:** Not applicable.

**Informed Consent Statement:** Not applicable.

**Data Availability Statement:** Not applicable.

**Conflicts of Interest:** The authors declare no conflict of interest.

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
