# Peer review of "University–Industry Technology Transfer: Empirical Findings from Chinese Industrial Firms"

_sustainability, doi:10.3390/su14159582_

Round 1
Reviewer 1 Report
1. there is insufficient review of relevant research progress in the introduction part.
2. In terms of the methods, the part of Geographic technology transfer and Dynamics of Chinese university–industry technology transfer network is just the way the results are presented and should not be listed as a method.
3. In terms of the results, you mentioned the license contracts collected between 2009 and 2014. I am not sure which year's results are presented in the part of the Overview of full network representing Chinese university-firm knowledge transfer. Besides, the Geographically mapped in the Figure6 is just up to 2013. Based on which year were the BGRM and BiRank centrality calculated? Moreover, the Chinese map omitted Taiwan,it is one part of China. What's wrong with the table3 and table 4, all ten columns are the same. The analysis for the results is relatively superficial and does not reveal some intrinsic mechanism。
4. On the basis of this empirical analysis, possible recommendations can be provided.
Reviewer 2 Report
Dear Authors,
The presented research topic “University–industry technology transfer: empirical findings from Chinese industrial firms” aims to study on Chinese university–industry knowledge transfer using a bipartite social network analysis (SNA) method, which emphasizes centrality estimates.
Abstract:
The abstract should be revised according to the guidelines of the journal.
Keywords: keywords should be ordered according to the journal's instructions, e.g. research area is placed last.
After reading the paper, I have comments and suggestions to improve the paper as follows:
In INTRODUCTION
This chapter should state the research problem and the research hypotheses or questions.
[113-116] - I suggest removing this piece of text. There is no need to write what each chapter will be about.
Chapter missing or LITERATURE REVIEW and THEORETICAL BACKGROUND
In DATA AND METHODS
This chapter is well presented.
I suggest to present a scheme of research procedure.
The Results
The results are presented and described in a good way and are very interesting.
In the Discussion Section, the authors should discuss and explain the findings and results of the paper more. This would contribute to a high improvement of this paper. The authors should compare their project and results with results from similar conducted research on this topic from other parts of Europa and all around the world.
The chapter should still answer the question: what tangible benefits this study has brought to the development of innovation, and make recommendations.
Technical errors to be corrected:
There should be numbers in the text instead of the author's name.
The literature list needs to be improved according to the journal's guidelines.
Kind regards,
Reviewer
Round 2
Reviewer 1 Report
There has great improvements for this paper.
It can be accepted after double-checking. The number of decimal places should better be consistent
Author Response
Thank you for your comments concerning our manuscript.
We have checked our manuscript carefully.
And we have made number of decimal places consistent, except those difference can only be observed from multiple digits after the decimal point.
We hope this revision can meet with approval.